# Burnout, Resilience, Supervisory Support, and Quitting Intention among Healthcare Professionals in Saudi Arabia: A National Cross-Sectional Survey

**DOI:** 10.3390/ijerph20032407

**Published:** 2023-01-29

**Authors:** Rayan A. Siraj, Ahmed E. Alhaykan, Ahmed M. Alrajeh, Abdulelah M. Aldhahir, Jaber S. Alqahtani, Samah Bakhadlq, Saeed M. Alghamdi, Abdullah A. Alqarni, Manal M. Alqarni, Turki M. Alanazi, Abdullah Alruwaili, Saleh S. Algarni, Fahad H. Alahmadi, Mushabbab Alahmari, Rashid H. Alahmadi

**Affiliations:** 1Department of Respiratory Care, College of Applied Medical Sciences, King Faisal University, Al-Ahasa 31982, Saudi Arabia; 2Respiratory Therapy Department, Faculty of Applied Medical Sciences, Jazan University, Jazan 45142, Saudi Arabia; 3Department of Respiratory Care, Prince Sultan Military College of Health Sciences, Dammam 34313, Saudi Arabia; 4Clinical Technology Department, Respiratory Care Program, Faculty of Applied Medical Sciences, Umm Al-Qura University, Makkah 21961, Saudi Arabia; 5Department of Respiratory Therapy, Faculty of Medical Rehabilitation Sciences, King Abdulaziz University, Jeddah 22254, Saudi Arabia; 6Department of Respiratory Therapy, King Saud Bin Abdelaziz University for Health Sciences, Al Ahsa 31982, Saudi Arabia; 7King Abdullah International Medical Research Center, Al Ahsa 31982, Saudi Arabia; 8Emergency Medical Services Program, College of Applied Medical Sciences, King Saud Bin Abdulaziz University for Health Sciences, Al Ahsa 31982, Saudi Arabia; 9Department of Respiratory Therapy, College of Applied Medical Sciences, King Saud Bin Abdulaziz University for Health Sciences, Riyadh 12271, Saudi Arabia; 10King Abdullah International Medical Research Center, Riyadh 12271, Saudi Arabia; 11Respiratory Therapy Department, College of Medical Rehabilitation Sciences, Taibah University, Madinah 41411, Saudi Arabia; 12Department of Respiratory Therapy, Faculty of Applied Medical Sciences, University of Bisha, Bisha 67114, Saudi Arabia; 13Taibah Primary Health Centre, Ministry of Health, Madinah 42353, Saudi Arabia

**Keywords:** burnout, resilience, supervisor support, healthcare

## Abstract

Although personal resilience and supervisory support are known to reduce the impact of burnout and quitting intention, there is limited data available to explore these relationships among healthcare professionals (HCPs) in Saudi Arabia. This study aimed to assess the prevalence of burnout and explore its association with resilience, supervisory support, and intention to quit among Saudi Arabian HCPs. Methods: A cross-sectional survey was distributed to a convenience sample of HCPs between April and November 2022. Participants responded to socio-demographic questions, the Maslach Burnout Inventory-Human Services Survey for Medical Personnel (MBI-HSS (MP)), the Connor-Davidson resilience scale 10 (CD-RISC 10), and the Perceived of Supervisor Support Scale (PSS). Descriptive, inferential, correlation, and logistic regression tests were performed for data analyses. Results: Of the 1174 HCPs included in the analysis, 77% were presented with high burnout levels: 58% with emotional exhaustion (EE), 72% with depersonalization (DP), and 66% with low personal accomplishment (PA). Females were associated with increased odds of burnout (OR: 1.47; 95% CI: 1.04–2.06) compared to males. Burnout and its subscales were associated with higher intention to leave practice, with 33% of HCPs considering quitting their jobs. Furthermore, HCPs reported a low resilience score overall, and negative correlations were found between EE (r = −0.21; *p* < 0.001) and DP (r = −0.12; *p* < 0.01), and positive correlation with low PA (r = 0.38; *p* < 0.001). In addition, most HCPs perceived supervisory support as low, and it is associated with increased burnout and quitting intention. Conclusion: Burnout is common among HCPs across all clinical settings and is associated with higher intention to quit and low resilience and supervisory support. Workplace management should provide a supportive workplace to reduce burnout symptoms and promote resiliency.

## 1. Introduction

Burnout syndrome is a public health epidemic that encompasses three dimensions: emotional exhaustion (EE), depersonalization (DP), and reduced personal accomplishment (PA) [1]. It is a global concern for healthcare professionals (HCPs) [2], which occurs due to extended work stress and has not been managed optimally. In addition to the significant impact of burnout on mental and physical health [3], it is also associated with increased medical error incidents [4], reduced patient safety [5], improper delivery of healthcare services [6], frequent absence from work, increased rates of staff resignation, and even the intention to leave the profession [7].

Increasing evidence suggests that burnout is highly prevalent among HCPs. However, estimates of burnout prevalence vary largely, depending on the sample population, clinical settings, and profession. For instance, a previous cross-sectional study conducted in the Southern region of Saudi Arabia, which included 95 physicians and 187 nurses working in different emergency departments, indicated that 88% of HCPs experienced emotional exhaustion. Another study, which included 66 respiratory therapists (RTs) working in a main tertiary hospital in the capital of Saudi Arabia, revealed that RTs experienced a significant level of burnout across the three dimensions: 77% with EE, 98% with DP, and 73% with low PA [8]. The exact cause of increased burnout is unknown, but factors such as years of experience, working in critical settings, staff shortage, and management support have been suggested. 

Resilience, a multi-dimensional construct that reflects the individual’s ability to cope with stressful events, is a factor that positively impacts overall health and can mitigate and protect against burnout [9] Highly resilient HCPs are more likely to adapt to stressful situations and high workloads in clinical settings [10,11] thus, they are less likely to be affected by burnout syndrome. Indeed, several studies conducted across different healthcare disciplines and settings showed that higher resilience is associated with lower burnout levels [12,13] and also correlates negatively with EE and DP and positively with PA [12]. Therefore, promoting resiliency among HCPs should be prioritized. 

Supervisory support, the extent to which supervisors and managers care for their employees, is a valuable resource that can stimulate personal growth and reduce the impact of burnout. Current evidence suggests that supervisory support and burnout are related. In a study of 203 nurses study, Kalliath et al. (2002) showed that low supervisory support was linked to high burnout levels and even the intention to leave the practice [14]. Further, a recent longitudinal study that included 195 HCPs showed that supervisory support was significantly associated with lower emotional exhaustion and higher job satisfaction [15]. It seems likely that the amount of support HCPs receive is a key factor in reducing symptoms of burnout and quitting intention. Thus, increasing supervisory support should be routinely practiced in clinical settings. 

Healthcare professionals are prone to increased burnout levels across different clinical venues. Understanding the relationships between HCPs’ burnout, resilience, supervisory support, and quitting intention is vital for healthcare workers and patient outcomes. Thus, we sought to (1) assess the national prevalence of burnout and its dimensions (EE, DP, and PA), and (2) explore the relationships between burnout and resilience, supervisory support, and quitting intention among Saudi Arabian HCPs. 

This study aimed to answer the following research questions: (1) what is the prevalence of burnout and its subscales among HCPs in Saudi Arabia? and (2) are there associations between burnout with resilience, supervisory support, and quitting intention in a national sample of Saudi HCPs? 

### Theoretical Framework

The burnout phenomenon can be explained by two theories: the Job Demands–Resources (JD-R) model [16] and the Conservation of Job Resources (COR) theory [17]. The JD-R subdivides the risk factors associated with job stress and burnout into job demands and resources. Environmental stressors such as significant workload, extended working hours, and frequently changing shift work are significant job demands which eventually contribute to an increased risk of emotional exhaustion [16,18]. Job resources such as feedback, job security, autonomy, and supervisor support [18] are essential for meeting organizational goals, promoting personal growth, and minimizing the impact of job demands. Indeed, high job demands, such as that encountered by HCPs, and low resources are important predictors of burnout. In addition, the COR theory states that burnout occurs due to several reasons, such as threatened or lost resources or individuals’ failure to gain needed resources. Thus, job recourses are vital in achieving other valued resources [17].

Supervisory support, represented by the Organizational Support Theory (OST), is an important job resource that buffers the effect of job stress and burnout [18]. It shows how much organizations value the contributions and well-being of their employees. In other words, employees, from their perspectives, consider the actions taken by their supervisors to be connected with organizational activities. The OST supports a self-enhancement process that leads to identification with the organization, affective organizational commitment, and building strong connections between employees and their superiors at the organizational level. This will eventually result in positive outcomes such as increased job satisfaction, improved work performance, and reduced stress, including burnout.

The conceptual framework of this paper helps in understating the associations between burnout, resilience, supervisory support, and quitting intention in the healthcare context by linking three concepts: COR theory, JD-R model, and OST. It explains how high job demands and low resources contribute to increased burnout, and the role of job resources, such as supervisory support, in mitigating the impacts of burnout. Further, it also shows that lost or threatened resources may interfere with the capabilities of HCPs to be resilient to cope with the negative impacts of environmental demands. The OST provides explanations for the nature of supervisory support and its positive outcomes on employee commitment, performance, and satisfaction within the organization. Increased job satisfaction as a result of high levels of supervisory support might lead to a decrease in the intention to leave the job. 

## 2. Methods

### 2.1. Study Design

The current study’s design was cross-sectional, using an online questionnaire to assess the prevalence of burnout and its relationships with resilience, supervisory support, and intention to quit among HCPs. An electronic platform (Survey Monkey) was used for the data collection from 1 April 2022 to 30 November 2022. 

### 2.2. Participants, Sampling Strategy and Data Collection

One thousand three hundred healthcare providers working in different clinical settings across all regions in Saudi Arabia were approached using a non-probability convenience sampling strategy. HCPs were included in the final analyses if they were 18 years and older, licensed to practice by the Saudi Commission for Health Specialties (SCFHS), and, most importantly, officially agreed to participate in the study. A cover page was included at the beginning of the questionnaire, explaining the study’s aim and the principal investigator’s identity. It was made clear that taking part in the survey was 100% voluntary and that informed consent had to be obtained before the study. The cover page also included information on the confidentiality of the collected data and that there would only be used to research purposes. No personal information or identification was collected. To reach the target population, social networks (Twitter, WhatsApp, and Telegram) were used to distribute the survey. After participants had filled out the questionnaires and submitted them to the web server, data was transferred into an excel sheet for future use. The estimated time to complete the questionnaire ranged from 5–7 min. 

### 2.3. Instruments

The instrument package included socio-demographic questions, Maslach Burnout Inventory–Human Services Survey for Medical Personnel (MBI–HSS (MP)) [19], Connor-Davidson resilience scale 10 (CD-RISC 10), and Perception of Supervisory Support Scale (PSS). The socio-demographic variables are age, gender, profession, region, working sector, working hospital, years of experience, living arrangement, marital status, income, current shift (day/night), and quitting intention.

### 2.4. Burnout

Burnout was measured using the MBI–HSS (MP) questionnaire. The questionnaire comprises 22 questions and measures three independent subscales: (1) emotional exhaustion (EE) with nine questions; (2) depersonalization (DP) with five questions; and (3) personal achievement (PA) with eight questions. A seven-point Likert Scale was used to assess the frequency of each burnout dimension (EE, DP, and PA) experienced by HCPs at their workplace. The range of each individual response was from 0 (never) to 6 (every day). The total calculated scores were then used to determine whether HCPs were classified as having low, moderate, or high burnout in each burnout dimension. A high level of burnout was indicated if HCPs achieved high scores of EE and DP and a low score of PA [20]. 

Measuring the prevalence of burnout subscales was determined using the sum methods. The sum method consists of adding the responses of each item within each individual dimension and calculating the total score for that dimension. The cut-off scores for high burnout were determined as follows: (1) EE of ≥ 27; (2) DP of ≥ 10; and (3) PA of ≤33. The cut-off scores for moderate burnout were determined as follows: (1) EE between 19–26; (2) DP between 6–9; and (3) PA between 34–39. The cut-off score for determining low burnout was as follows: EE of ≤ 18; 2) DP of ≤ 5; and 3) PA of ≥ 40 [20].

The MBI–HSS (MP) is a valid and reliable tool and has been widely used in the literature. Cronbach’s alpha was used to estimate the internal consistency and the results showed high consistency (measured with Cronbach’s Alpha) with 0.90 for EE, 0.79 for DP, and 0.71 for PA [19], suggesting that the instrument is intended to measure what is supposed to be measured burnout in this case. 

### 2.5. Resilience

HCPs’ resilience was measured using the Connor–Davidson resilience scale 10 (CD-RISC 10). The CD-RISC 10 consists of a ten 5-point Likert scale, ranging from 0 (not true at all) to 4 (true nearly all the time). The total score ranged from 0–40, and a higher score indicates higher self-reported resilience [21]. The questionnaire has been used widely across healthcare professions, with Cronbach’s alpha being 0.96 [22].

### 2.6. Perceived Supervisory Support Scale (PSS)

Supervisory support was measured using adapted items, which are intended to measure organization support, consisting of 8 items. For the purpose of this study, the word “organization” was replaced with “supervisor”. Each item was measured by a 7-point Likert scale ranging from 0 (strongly disagree) to 7 (strongly agree). The questionnaire intends to assess the level of support experienced by HCPs. The responses to the 8 items included in the survey were calculated, and the mean indicates whether HCPs perceived their supervisory support as low, medium, or high. The cut-off points for the three categories were determined as follows: a mean score of 1–2.9 indicated low support, 3.0–5.0 indicated medium support, and 5.1–7.0 indicated high support. The internal reliability of the questionnaire ranges from 0.90 to 0.93. Scoring of the SPSS [23].

### 2.7. Ethical Consideration

This study sought ethical approval from an independent research committee at King Faisal University, Saudi Arabia (ID: KFU-REC-2022-APR-EA000563). 

### 2.8. Statistical Analysis

STATA version 16.0 software StataCorp LP, College Station, TX, USA) was used for statistical analyses and data management. The baseline characteristics of the study participants were described in terms of frequency or mean (SD) as appropriate. The normality of continuous variables was assessed using histograms to determine the appropriate statistical tests. An Independent sample t-test was used to compare the mean difference between independent groups (e.g., the mean difference of EE between HCPs with quitting intention vs. without). Pearson’s correlation was performed to assess the correlation of resilience and supervisory support with burnout subscales (EE, DP, and PA. Logistic regression models were also performed to determine the factors associated with burnout and investigate the association between burnout with quitting intention and supervisory support. 

## 3. Results

Overall, 1300 HCWs showed interest in participating in the study. Of those, 1174 participants completed the online questionnaires and were included in the final analyses, Figure 1. Males accounted for 54.4%, and the mean (SD) age of HCWs was 31 (6.2) years. Nurses accounted for 37.8%, followed by RTs and physicians, 26.8% and 26.7%, respectively. Most HCWs worked in the governmental sector (87.5%) and had between 1–4 years of clinical experience (40.6%). Interestingly, a third of the participants have considered leaving their jobs, and 25.6% had been prescribed or diagnosed with mental health issues within the past three months (Table 1). 

### 3.1. Prevalence of Burnout

Moderate to high burnout levels were presented in 87% of the HCPs. The mean (SD) scores for the burnout subscales were as follows: EE of 27.6 (11.8), DP of 13.8 (7.6), and PA of 28 (10.8). In addition, the prevalence of the burnout subscales among HCPs was as follows: 58.4% with high emotional exhaustion, 72.4% with high depersonalization, and 66% with low personal accomplishment (Table 2).

### 3.2. Factors Associated with Burnout among HCWs

Using logistic regression analyses, we aimed to assess the level of independent associations of demographic and workplace variables with burnout. Being a female was associated with increased burnout compared to males (OR: 1.47; 95% CI: 1.04–2.06). However, being married and having more clinical experience (>10 years) were protective against high burnout. In addition, earning more than 20,000 SR/month was associated with less burnout, despite not reaching a statistical significance (Table 3). 

We also investigated the association between burnout and its subscales with the intention to leave the job. An independent sample t-test showed that HCWs who had considered quitting their jobs had a high score of burnout (mean (SD) = 75.5 (19.2) compared to those who had not considered leaving their jobs (mean (SD) = 66.54 (28.67)), *p* < 0.001. Similarly, EE, DP, and low PA scores were also greater among HCWs with quitting intention, *p* < 0.005. Moreover, HCWs who had considered leaving their jobs were at increased risk of burnout (OR: 6.3; 95% CI: 3.54–11.35), EE (OR: 4.48; 95% CI: 2.97–6.75), DP (OR: 2.91; 95% CI: 1.96–4.33), and low PA (2.71; 95% CI: 1.85–3.98) compared to those who did not have the intention to quit. 

### 3.3. Associations between Resilience, Burnout, and Intention to Quit

Resilience was assessed using Connor–Davidson Resilience Scale (CD-RISC 10). The mean (SD) resilience score was 27.2 (6.2), indicating a low overall resilience level. A significant association between burnout and resilience scores were observed, with those experiencing high/moderate burnout having lower levels of resilience (mean = 24.5, SD 5.8) compared to those with low burnout (mean = 27.6, SD 6.1), *p* < 0.001. Similarly, HCWs with high EE, DP, and low PA also showed low resilience scores compared to those with low EE, DP, and increased PA; (mean (SD) = 26.1 (6.8) vs. 27.5 (5.9), *p* < 0.001), (mean (SD) = 25.5 (6.1) vs. 27.5 (6.1), *p* < 0.001), and (mean (SD) = 31.9 (4.7) vs. 26.2 (5.9), *p* < 0.001), respectively. 

HCWs who had considered leaving their jobs showed low resilience scores (mean 26.1, SD 6.2) compared to those who did not have the intention to quit their jobs (mean 27.8, SD 5.8), *p* < 0.001. In addition, resilience-burnout subscales correlations showed negative correlations with EE (r = −0.21; *p* < 0.001) and DP (r = −0.12; *p* < 0.01) and positive correlations with low PA (r = 0.38; *p* < 0.001).

### 3.4. Associations between Burnout and its Subscales with Supervisory Support

HCWs had a mean (SD) score of PSS of 2.81 (0.68). Most HCWs (*n* = 556; 47.4%) indicated that they had low supervisory support, whilst only 12.3% of HCWs perceived a high level of supervisory support, Table 4. 

There were negative correlations between the perception of supervisory support and levels of burnout (r = −0.23; *p* < 0.001). In addition, supervisory support was negatively associated with EE (r = 0.28; *p* < 0.001), DP (r = -0.16; *p* = 0.03), and positively correlated with PA (r = 0.18; *p* < 0.002). Logistic regression analysis showed that high supervisor support was associated with lower burnout (OR: 0.53; 95% CI: 0.30–0.95) and intention to quit the job (OR: 0.16; 95% CI: 0.15–0.27). 

## 4. Discussion

Our study is the first to explore the relationship between burnout with resilience, supervisory support, and quitting intention among HCPs working in Saudi Arabia. The present study showed that HCPs suffer from increased burnout, which is significantly associated with a greater quitting intention. Furthermore, HCPs had overall low levels of resilience and perceived supervisory support; both were significantly correlated with increased burnout and its subscales. 

Burnout syndrome poses a significant issue among HCPs, leading to adverse outcomes at personal and clinical levels. Previous national studies revealed that the prevalence of burnout among doctors, nurses, and RTs is high (up to 98%) [8]. In concordance with this, we found that most HCPs included in our study (87%) experienced moderate to high levels of burnout. Indeed, HCPs, regardless of their professions, are exposed to numerous stressors such as long working hours, high workload (clinical and administrative), insufficient compensation, and lack of clinical autonomy, all of which contribute to increased stress levels and untimely burnout. 

Despite that HCPs (irrespective of gender) are exposed to similar stressors in clinical settings, this study showed that females were more at increased risk of burnout symptoms compared to males. In line with our findings, a previous systemic review, including 138 articles on middle eastern HCPs, showed that female HCPs are more likely to experience burnout symptoms than males [24]. Female physicians, for instance, have been shown to report mental health issues and even suicidal thoughts more than male physicians. The reasons for this are not clearly understood but might be attributed to female predominance in patient-facing roles, gender-related discrimination, gendered expectations in providing care, sexual harassment, and inequities [25]. There is also the lack of attention to “dual shift” work with high workloads at home, as women spend more time on unpaid work per day compared with men as parents and primary caregivers to family members.

Our findings also showed that being married as well as having long experience in the field were associated with low burnout. Family support is recognized as an important protective factor against burnout. A previous study revealed that 78% of nurses and 91% of physicians consider family support the most beneficial personal resource to protect against burnout [26]. Family support has also been inversely correlated with emotional exhaustion and depersonalization [27]. Moreover, our study showed that longer working years (>10 years) was independently associated with a lower burnout rate. Although this may seem unrealistic at first glance, several reasons could explain this observation. As years go by, HCPs become more competent in their jobs, more mature when they face certain situations, and that they have gained enough experience in life; all of which contribute to a lower level of emotional exhaustion, and thus, burnout [28,29]. 

Increasing evidence suggests that an increased level of burnout is a well-known driver of intention to leave the practice. In line with this, our analysis showed that one-third of HCPs had considered leaving the practice, and those with quitting intention were more likely to experience increased burnout and lower resilience and support from their supervisors. These findings are supported by the fact that HCPs are prone to significant burnout related to their job demands, such as workload, low work control, and high psychologic demands are related to burnout. Current literature shows that 33% of nurses had the intention to leave their jobs within the next year [30,31]. Moreover, several studies (among physicians and nurses) also support the association between quitting intention and burnout (irrespective of the clinical settings), with ORs ranging from 2 to 5, similar to the findings of this study [32,33]. The main reasons for considering leaving the practice are physical, mental, workloads, and job stresses, which highlight the need to develop strategies to improve the workplace environment and HCPs retention. 

Current literature has highlighted the role of resilience in terms of facilitating healthy and more productive workplace [34,35]. Indeed, highly resilient HCPs are likely to deal more effectively with stressful events such as those encountered across different clinical settings and heavy workloads [10,11]. In this study, the mean score of resiliency was relatively low and only a small proportion reached the standards of high resilience. Given the high prevalence of burnout reported in this study, it is nevertheless expected that HCPs would be less resilient. In addition, resilience was also associated with all three subscales of burnout (EE, DP, and PA). These findings are consistent with current literature, as higher resilience is linked to low burnout. Thus, there need to be strategies implemented at the workplace to reduce the impact of burnout and promote resiliency among HCPs. 

Supervisory support is an important resource that has been linked to ameliorating the impact of burnout symptoms and stimulating personal development among different healthcare disciplines. [36]. In this study, perceived supervisory support was negatively associated with burnout, and low supervisory support was associated with increased odds of quitting intention. In concordance with our findings, a 12-month longitudinal study, which included 195 healthcare practitioners, showed that high supervisory support was associated with reduced quitting intention through mitigating burnout symptoms [15]. This implies that increasing supervisory support, as an intervention in routine practices, is crucial in ameliorating burnout symptoms to reduce leaving intention. 

## 5. Strengths and Limitations

A major strength of this study is that it is the first study to assess the relationship between burnout with resilience, supervisory support, and quitting intention among HCPs at the national level; something that has not been explored before. In addition, HCPs were recruited from various clinical venues and all regions of Saudi Arabia, thus, offering high external validity. However, this study has some limitations. First, the cross-sectional nature of this study does not allow for the establishment of causality. Second, since most HCPs had more than one year of clinical experience, the effect of COVID-19 on burnout cannot be excluded. In addition, we could not assess the perception of those with access to training and pre-employment programs, which could help reduce burnout and promote resilience.

## 6. Conclusions

Burnout and its subscales (EE, DP, and PA) were highly prevalent among healthcare workers and were associated with an increased risk of leaving practice. Low resilience and supervisory support were associated with increased burnout. There need to be effective interventions to promote resiliency. Supervisors should also provide a supportive work environment to mitigate burnout symptoms at the workplace.

## Figures and Tables

**Figure 1 ijerph-20-02407-f001:**
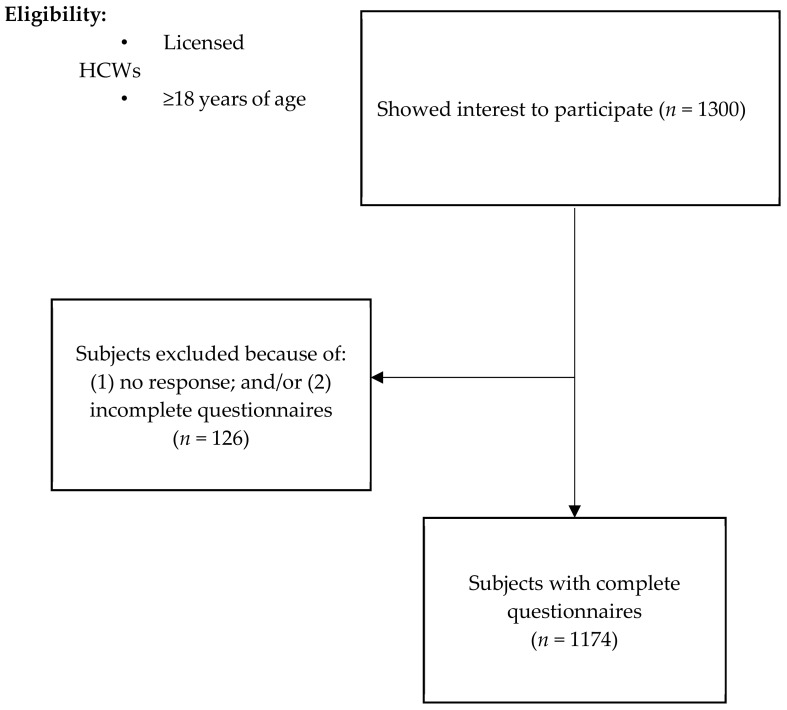
Participants’ selection to the study.

**Table 1 ijerph-20-02407-t001:** Characteristics of study participants (*n* = 1174).

Variable	
Age, years (mean (SD))	31 (6.2)
Gender (male %)	639 (54.4%)
Profession, *n* (%)	
Physicians	314 (26.7%)
Nurse	444 (37.8%)
Respiratory Therapists	315 (26.8%)
Others	101 (8.7%)
Working Hospital, *n* (%)	
Medical City	361 (30.8%)
Military Hospital	207 (17.6%)
Primary Care Clinic	100 (8.5%)
Specialized Hospital	160 (13.6%)
University Hospital	70 (6%)
General Hospital	228 (19.4%)
Psychiatry Hospital	48 (4.1%)
Sector, *n* (%)	
Governmental	1027 (87.5%)
Private	147 (12.5%)
Geographical Region, *n* (%)	
Eastern	109 (9.3%)
Central	226 (19.2%)
Western	298 (25.4%)
Southern	393 (33.5%)
Northern	148 (12.6%)
Working Settings, *n* (%)	
General Wards	258 (22%)
ER	105 (9%)
ICUs	317 (27%)
OR	151 (13%)
Recovery/Post-op	129 (11%)
Rehabilitation Center	70 (8%)
General Clinic	46 (4%)
Specialized Clinic	35 (3%)
Training Center	23 (2%)
Others	11 (1%)
Years of clinical experience, *n* (%)	
<1 year	203 (17.3%)
1–4 years	447 (40.6%)
5–10 years	133 (11.3%)
>10 years	203 (17.3%)
Living arrangements, *n* (%)	
Living alone	183 (24%)
Living with family	784 (66.8%)
Living elsewhere	107 (9.2%)
Marital status, *n* (%)	
Single	497 (42.3%)
Married	636 (54.2%)
Divorced/Separated/Widowed	41 (3.5%)
Monthly income in SR, *n* (%)	
<12,000	446 (38%)
12,000–20,000	662 (53%)
>20,000	106 (9%)
Current shift work, *n* (%)	
Day	76 (40%)
Night	114 (60%)
Intention to leave the job, *n* (%)	
Yes	390 (33.2%)
No	784 (66.8%)

Data are presented as frequency (%) or mean SD. Abbreviation: ER: emergency room; ICU: intensive care unit; OR: operation room; SR: Saudi Riyal.

**Table 2 ijerph-20-02407-t002:** Prevalence of burnout among HCWs in Saudi Arabia (*n* = 1174).

Burnout	
High	909 (77.4%)
Moderate	111 (9.5%)
low	154 (13.1%)
Burnout subscale	
Emotional exhaustion	
High	686 (58.4%)
Moderate	251 (21.4%)
Low	237 (20.2%)
Depersonalization	
High	850 (72.4%)
Moderate	124 (10.6%)
Low	200 (17%)
Personal accomplishment	
Low	775 (66%)
Moderate	193 (16.4%)
High	206 (17.6%)

Results are presented as frequency (%) unless stated otherwise.

**Table 3 ijerph-20-02407-t003:** Demographic and workplace factors associated with increased burnout.

Variables	OR Burnout	95% CI
Gender		
Male	Reference	Reference
Female	1.47	1.04–2.06
Marital status		
Single	Reference	Reference
Married	0.28	0.13–0.58
Divorced/widowed	0.71	0.49–1.02
Years of clinical experience		
<1 year	References	References
1–4 years	1.2	0.65–2.26
5–9 years	0.91	0. 54–1.5
>10 years	0.17	0.10–0.30
Monthly income (SR)		
<20,000 SR	Reference	Reference
>20,000 SR	0.90	0.51–1.41

Results are presented as frequency (%) unless stated otherwise. Abbreviation: SR: Saudi Riyal; OR: odds ratio.

**Table 4 ijerph-20-02407-t004:** Supervisory Support by Levels.

PSS, mean (SD) **PSS levels, *n* (%)** High (5.1–7.0) Medium (3.0–5.0) Low (0.1–2.9)	2.81 (0.68) 144 (12.3%) 473 (40.3%) 556 (47.4%)

PSS: perceived supervisory support.

## Data Availability

The data presented in this study are available on reasonable requestfrom the corresponding author.

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
