# Peer review of "Burnout, Resilience, Supervisory Support, and Quitting Intention among Healthcare Professionals in Saudi Arabia: A National Cross-Sectional Survey"

_ijerph, 2023, doi:10.3390/ijerph20032407_

Round 1

Reviewer 1 Report

This is an interesting and much needed research in the context of Saudi Arabia on healthcare professionals. I have highlighted the following points and recommend you to address these points before you submit the revised version of manuscript.

1. I recommend you to transform the structured abstract into one comprehensive paragraph or if you want to keep the abstract structured, then you have to assign one short paragraph to each section such as purpose, design etc. 

2. I can see that in the fourth line of fourth paragraph of introduction 'Kalliath et al.' has no year of publication which brings in bad impression. 

3. I believe, there is a need to include one more section "literature review" having the detailed description of theoretical model of this study and having the theoretical support for the proposed model. 

4. Method section id fine and well written however, I will recommend you to add at least one sample item for every instrument in the instrument section and also add the Cronbach's Alpha score for every instrument. 

5. Results are presented well.

6. Discussion and conclusion sections are well written.  

Author Response

We greatly appreciate the reviews by the Reviewers. We have revised our paper in the light of the useful comments, and we hope we have addressed all concerns indicated in the review report.

Response to Comments from Reviewer 1

This is an interesting and much needed research in the context of Saudi Arabia on healthcare professionals. I have highlighted the following points and recommend you to address these points before you submit the revised version of manuscript.

Comment 1

I recommend you to transform the structured abstract into one comprehensive paragraph or if you want to keep the abstract structured, then you have to assign one short paragraph to each section such as purpose, design etc. 

Response 1:

Thank you for the comment. The abstract has now been re-structured in accordance with the reviewer comment. It now reads as follows “ Abstract: Although personal resilience and supervisory support are known to reduce the impact of burnout and quitting intention, there is limited data available to explore these relationships among healthcare professionals (HCPs) in Saudi Arabia.  This study aimed to assess the prevalence of burnout and explore its association with resilience, supervisory support, and intention to quit among Saudi Arabian HCPs. Methods: A cross-sectional survey was distributed to a convenience sample of HCPs between April and November 2022. Participants responded to socio-demographic questions, the Maslach Burnout Inventory-Human Services Survey for Medical Personnel (MBI-HSS (MP)), the Connor-Davidson resilience scale 10 (CD-RISC 10), and the Perceived of Supervisor Support Scale (PSS). Descriptive, inferential, correlation and logistic regression tests were performed for data analyses. Results: Of the 1,174 HCPs included in the analysis, 77% were presented with high burnout levels: 58% with emotional exhaustion (EE), 72% with depersonalization (DP), and 66% with low personal accomplishment (PA). Females were associated with increased odds of burnout (OR: 1.47; 95% CI: 1.04 – 2.06) compared to males. Burnout and its subscales were associated with higher intention to leave practice, with 33% of HCPs considering quitting their jobs. Furthermore, HCPs reported low resilience score overall, and negative correlations were found between with EE (r= - 0.21; p<0.001) and DP (r = - 0.12; p<0.01) and positive correlation with low PA (r = 0.38; p<0.001). In addition, most HCPs perceived supervisory support as low, and it is associated with increased burnout and quitting intention. Conclusion: Burnout is common among HCPs across all clinical settings and is associated with higher intention to quit and low resilience and supervisory support. Workplace management should provide a supportive workplace to reduce burnout symptoms and promote resiliency.”

Comment 2:

I can see that in the fourth line of fourth paragraph of introduction 'Kalliath et al.' has no year of publication which brings in bad impression. 

Response 2:

The year of Kalliath has been added. It now reads as follows (introduction; lines 85) “ In one study, Kalliath et al. (2002) showed that low supervisory support was linked to high burnout levels among nurses and intention to quit the job”.

Comment 3:

  1. I believe, there is a need to include one more section "literature review" having the detailed description of theoretical model of this study and having the theoretical support for the proposed model. 

Response3:

Thank you for the comment. We have added a suction on the theoretical framework (lines 105-137)” The burnout phenomenon can be explained by two theories: the Job Demands-Resources (JD-R) model (13)and the Conservation of Job Resources (COR) theory (14). The JD-R subdivides the risk factors associated with job stress and burnout into job demands and resources. Environmental stressors such as significant workload, extend-ed working hours, and frequently changing shift work are significant job demands which eventually contribute to an increased risk of emotional exhaustion (13, 15). Job resources such as feedback, job security, autonomy, and supervisor support (15) are es-sential for meeting organisational goals, promoting personal growth, and minimizing the impact of job demands. Indeed, high job demands, such as that encountered by HCPs, and low resources are important predictors of burnout.  In addition, the COR theory states that burnout occurs due to several reasons, such as threatened or lost re-sources or individuals’ failure to gain needed resources. Thus, job recourses are vital in achieving other valued resources (14).

Supervisory support, represented by the Organizational Support Theory (OST), is an important job resource that buffers the effect of job stress and burnout (15). It shows how much organizations value the contributions and well-being of their employees. In other words, employees, from their perspectives, consider the actions taken by their supervisors to be connected with organizational activities. The OST supports a self-enhancement process that leads to identification with the organization, affective or-ganizational commitment, and building strong connections between employees and their superiors at the organizational level. This will eventually result in positive out-comes such as increased job satisfaction, improved work performance and reduced stress, including burnout.

The conceptual framework of this paper helps in understating the associations between burnout, resilience, supervisory support, and quitting intention in the healthcare context by linking three concepts: COR theory, JD-R model and OST. It explains how high job demands and low resources contribute to increased burnout, and the role of job resources, such as supervisory support, in mitigating the impacts of burnout. Further, it also shows that lost, or threatened resources may interfere with the capabilities of HCPs to be resilient to cope with the negative impacts of environmental demands. The OST provides explanations for the nature of supervisory support, and its positive outcomes on employee commitment, performance, and satisfaction within the organization. Increased job satisfaction as a result of high levels of su-pervisory support might lead to a decrease in the intention to leave the job. 

Comment 4:

Method section id fine and well written however, I will recommend you to add at least one sample item for every instrument in the instrument section and also add the Cronbach's Alpha score for every instrument. 

Response 4:

Thank you very much. This has now been added.

Comment 5:

Results are presented well.

Response 5:

Thank you.

Comment 6:

Discussion and conclusion sections are well written. 

Response 6: 

Thank you.

Reviewer 2 Report

I would like to thank you for the opportunity since I feel very fortunate to be able to review this article and I would like to congratulate the authors for this work. For me this topic is very important and has a lot of value. My suggestions are detailed below and my consideration at the end.

This manuscript investigated the national prevalence of burnout (EE, PD, and PA) and examined the association between burnout and resilience, supervisory support, and intention to quit among Saudi Arabian HCPs.

Title: The title is concrete, representative and indicative of the problem investigated in the manuscript and provides information about where the research was conducted and the subject group Congratulations!

Abstract: The abstract is clear and complies with the general rules for writing a good abstract. However, I would like to see a better description of the sample, indicating the context.  This is the most important section of the paper since it will be read many more times than even the manuscript itself, so it needs the most attention. A brief note on the importance of the research is an excellent ending to a high-level abstract.

Introduction

As I mentioned, I find this research extremely important in contributing to the field of health care. I do not disagree with the authors' justifications and read many very good and current arguments.  However, it is suggested to provide more information about the conditions that must be present to consider that a person suffers from burnout and the conditions that must be present to consider that a person has good or bad resilience.

It is suggested to the authors that, based on the stated objective, they highlight the research questions that help to conduct the research and the discussion based on the findings in which the study variables, the study population, and the expected result appear.

Material and method.

Design and setting of the study. It is suggested to include information about the justification for the choice of an electronic tool to administer the questionnaires.

Instruments: Were the instruments used factorial? Information should be provided on the population with which the instruments used were validated. Was any type of adaptation to the study population used?

Participants. This section should be better defined. In this section (participants), the characteristics of the sample should be included. The number of participants, characteristics, inclusion-exclusion criteria, etc., are not reported. 

Statistical analysis. It is stated that the principle of normality was evaluated, the test used should be reported. 

Results: 

The results are correctly displayed and are easy to read and simple for a scholar not used to quantitative methodology. 

Discussion: It seems to me that a great job has been done in comparing the findings with other studies. Congratulations. It is suggested to include a section on practical and theoretical implications to evaluate the scope of the research.

Conclusions: They are clear and provide an answer to the stated objectives. 

Author Response

Response to Comments from Reviewer 2

I would like to thank you for the opportunity since I feel very fortunate to be able to review this article and I would like to congratulate the authors for this work. For me this topic is very important and has a lot of value. My suggestions are detailed below and my consideration at the end.

This manuscript investigated the national prevalence of burnout (EE, PD, and PA) and examined the association between burnout and resilience, supervisory support, and intention to quit among Saudi Arabian HCPs.

Title: The title is concrete, representative and indicative of the problem investigated in the manuscript and provides information about where the research was conducted and the subject group Congratulations!

Comment 1:

Abstract: The abstract is clear and complies with the general rules for writing a good abstract. However, I would like to see a better description of the sample, indicating the context.  This is the most important section of the paper since it will be read many more times than even the manuscript itself, so it needs the most attention. A brief note on the importance of the research is an excellent ending to a high-level abstract.

Response 1:

Thank you. The Abstract has been revised in accordance with the reviewer suggestion. It now reads as follows “ Abstract: Although personal resilience and supervisory support are known to reduce the impact of burnout and quitting intention, there is limited data available to explore these relationships among healthcare professionals (HCPs) in Saudi Arabia.  This study aimed to assess the prevalence of burnout and explore its association with resilience, supervisory support, and intention to quit among Saudi Arabian HCPs. Methods: A cross-sectional survey was distributed to a convenience sample of HCPs between April and November 2022. Participants responded to socio-demographic questions, the Maslach Burnout Inventory-Human Services Survey for Medical Personnel (MBI-HSS (MP)), the Connor-Davidson resilience scale 10 (CD-RISC 10), and the Perceived of Supervisor Support Scale (PSS). Descriptive, inferential, correlation and logistic regression tests were performed for data analyses. Results: Of the 1,174 HCPs included in the analysis, 77% were presented with high burnout levels: 58% with emotional exhaustion (EE), 72% with depersonalization (DP), and 66% with low personal accomplishment (PA). Females were associated with increased odds of burnout (OR: 1.47; 95% CI: 1.04 – 2.06) compared to males. Burnout and its subscales were associated with higher intention to leave practice, with 33% of HCPs considering quitting their jobs. Furthermore, HCPs reported low resilience score overall, and negative correlations were found between with EE (r= - 0.21; p<0.001) and DP (r = - 0.12; p<0.01) and positive correlation with low PA (r = 0.38; p<0.001). In addition, most HCPs perceived supervisory support as low, and it is associated with increased burnout and quitting intention. Conclusion: Burnout is common among HCPs across all clinical settings and is associated with higher intention to quit and low resilience and supervisory support. Workplace management should provide a supportive workplace to reduce burnout symptoms and promote resiliency”.

Comment 2:

Introduction

As I mentioned, I find this research extremely important in contributing to the field of health care. I do not disagree with the authors' justifications and read many very good and current arguments.  However, it is suggested to provide more information about the conditions that must be present to consider that a person suffers from burnout and the conditions that must be present to consider that a person has good or bad resilience.

Response 2:

We appreciate the suggestion from the reviewer. In the introduction, we have discussed the impact of burnout on HCPs in terms of increased risk of physical and mental illnesses, poor delivery of healthcare services and even intention to quit. We also discussed that higher resilience is associated with low burnout and the importance of promoting resiliency. We have further provided more clarifications to our argument in the introduction, and we hope it is acceptable.

 Comment 3:

It is suggested to the authors that, based on the stated objective, they highlight the research questions that help to conduct the research and the discussion based on the findings in which the study variables, the study population, and the expected result appear.

Response 3:

Thank you. We revised our objective to the following “ Healthcare professionals are prone to increased burnout levels across different clinical venues. Understanding the relationships between HCPs' burnout, resilience, supervisory support, and quitting intention is vital for healthcare workers and patient outcomes. Thus, we sought to 1) assess the national prevalence of burnout and its dimensions (EE, DP, and PA), and 2) explore the relationships between burnout and resilience, supervisory support, and quitting intention among Saudi Arabian HCPs.

This study aimed to answer the following research questions: 1) what is the prevalence of burnout and its subscales among HCPs in Saudi Arabia? and 2) are there associations between burnout with resilience, supervisory support, and quitting intention in a national sample of Saudi HCPs?”

Comment 4:

Design and setting of the study. It is suggested to include information about the justification for the choice of an electronic tool to administer the questionnaires.

Response 4:

Thank you for the opportunity to clarify. The reason why electronic platform was chosen to distribute the questionnaire was due its simplicity to reach the target population. We have revised the previous sentence to the following one “  To reach the target population, social networks (Twitter, WhatsApp and Telegram) were used to distribute the survey”.

Comment 5:

Instruments: Were the instruments used factorial? Information should be provided on the population with which the instruments used were validated. Was any type of adaptation to the study population used?

Response 5:

Thank you. We acknowledged that all three data collection tools (burnout, resilience, and supervisory support) used in this study have been widely used previous research among similar populations, as mentioned in our methods. The items which were used to measure supervisory support were adopted from the items used to measure organization support. We have added the following sentences in the methods for clarity “ Supervisory support was measured using adapted items, which is intended to measure organization support, consisting of 8 items.  For the purpose of this study, the word "organization" was replaced with "supervisor".  

Comment 6:

Participants. This section should be better defined. In this section (participants), the characteristics of the sample should be included. The number of participants, characteristics, inclusion-exclusion criteria, etc., are not reported. 

Response 6:

We have added the following sentences to provide more clarifications “One thousand three hundred healthcare providers working in different clinical set-tings across all regions in Saudi Arabia were approached using a non-probability convenience sampling strategy. Potential participants were included in the study if they were aged 18 years and older, licensed to practice from the Saudi Commission for Health Specialties (SCFHS), and most importantly, agreed to participate”.

Comment 7:

Statistical analysis. It is stated that the principle of normality was evaluated, the test used should be reported. 

Response 7:

This has been addressed by adding the following sentence “ The normality of continuous variables was assessed using histogram to determine the appropriate statistical tests”.

Comment 8:

The results are correctly displayed and are easy to read and simple for a scholar not used to quantitative methodology. 

Response 8:

Thank you.

Comment 9:

Discussion: It seems to me that a great job has been done in comparing the findings with other studies. Congratulations. It is suggested to include a section on practical and theoretical implications to evaluate the scope of the research.

Response 9:

Thank you.

Comment 10:

Conclusions: They are clear and provide an answer to the stated objectives.

Response 10:

Thank you.

Round 2

Reviewer 2 Report

Dear authors, I am deeply grateful that my suggestions and appreciations have been taken into account. I consider that the article has been improved and can be published.